# Bcl10 Regulates Lipopolysaccharide-Induced Pro-Fibrotic Signaling in Bronchial Fibroblasts from Severe Asthma Patients

**DOI:** 10.3390/biomedicines10071716

**Published:** 2022-07-15

**Authors:** Rakhee K. Ramakrishnan, Khuloud Bajbouj, Maha Guimei, Surendra Singh Rawat, Zaina Kalaji, Mahmood Y. Hachim, Bassam Mahboub, Saleh M. Ibrahim, Rifat Hamoudi, Rabih Halwani, Qutayba Hamid

**Affiliations:** 1Sharjah Institute for Medical Research, College of Medicine, University of Sharjah, Sharjah P.O. Box 27272, United Arab Emirates; rramakrishnan@sharjah.ac.ae (R.K.R.); kbajbouj@sharjah.ac.ae (K.B.); u21105961@sharjah.ac.ae (Z.K.); bhmahboub@dha.gov.ae (B.M.); saleh.ibrahim@sharjah.ac.ae (S.M.I.); 2Department of Pathology, Faculty of Medicine, Alexandria University, Alexandria 21526, Egypt; maha.guimei@alexmed.edu.eg; 3College of Medicine, Mohammed Bin Rashid University, Dubai P.O. Box 505055, United Arab Emirates; surendrasingh.rawat@mbru.ac.ae (S.S.R.); mahmood.almashhadani@mbru.ac.ae (M.Y.H.); 4Rashid Hospital, Dubai Health Authority, Dubai P.O. Box 4545, United Arab Emirates; 5Lübeck Institute of Experimental Dermatology (LIED), University of Lübeck, 23562 Lübeck, Germany; 6Division of Surgery and Interventional Science, University College London, London WC1E 6BT, UK; 7Immunology Research Lab, College of Medicine, King Saud University, Riyadh P.O. Box 145111, Saudi Arabia; 8Meakins-Christie Laboratories, McGill University, Montreal, QC H3A 0G4, Canada

**Keywords:** severe asthma, bronchial fibroblasts, fibrosis, Bcl10, NF-κB pathway, cytokines

## Abstract

Subepithelial fibrosis is a characteristic hallmark of airway remodeling in asthma. Current asthma medications have limited efficacy in treating fibrosis, particularly in patients with severe asthma, necessitating a deeper understanding of the fibrotic mechanisms. The NF-κB pathway is key to airway inflammation in asthma, as it regulates the activity of multiple pro-inflammatory mediators that contribute to airway pathology. Bcl10 is a well-known upstream mediator of the NF-κB pathway that has been linked to fibrosis in other disease models. Therefore, we investigated Bcl10-mediated NF-κB activation as a potential pathway regulating fibrotic signaling in severe asthmatic fibroblasts. We demonstrate here the elevated protein expression of Bcl10 in bronchial fibroblasts and bronchial biopsies from severe asthmatic patients when compared to non-asthmatic individuals. Lipopolysaccharide (LPS) induced the increased expression of the pro-fibrotic cytokines IL-6, IL-8 and TGF-β1 in bronchial fibroblasts, and this induction was associated with the activation of Bcl10. Inhibition of the Bcl10-mediated NF-κB pathway using an IRAK1/4 selective inhibitor abrogated the pro-fibrotic signaling induced by LPS. Thus, our study indicates that Bcl10-mediated NF-κB activation signals increased pro-fibrotic cytokine expression in severe asthmatic airways. This reveals the therapeutic potential of targeting Bcl10 signaling in ameliorating inflammation and fibrosis, particularly in severe asthmatic individuals.

## 1. Introduction

Asthma is a chronic respiratory disease that is usually associated with inflammation and remodeling of the airways. Subepithelial fibrosis is a characteristic hallmark of airway remodeling in asthma, particularly in patients with severe asthma [1]. Unfortunately, current asthma medications cannot sufficiently target and ameliorate subepithelial fibrosis. Therefore, a deeper understanding of the fibrotic mechanisms in action is essential to identify novel therapeutic targets. This could aid in ameliorating fibrosis and improving the airway dynamics in asthma patients.

The Nuclear factor (NF)-kappaB (NF-κB) signaling pathway is a highly versatile pathway that plays a vital role in multiple biological processes, including cell growth, survival, development, immune response and inflammation [2]. NF-κB is a key signaling pathway in asthma pathogenesis, particularly in airway inflammation. Persistent activation of the NF-κB pathway was noted in peripheral blood mononuclear cells isolated from patients with severe uncontrolled asthma [3]. As a result, increased levels of pro-inflammatory mediators were secreted by these cells despite continuous systemic glucocorticoid treatment. At the same time, activation of NF-κB signaling has also been noted in lung fibroblasts during fibrogenesis [4]. Furthermore, NF-κB activation induces the expression of pro-fibrotic cytokines such as IL-6, IL-8 and TGF-β [5,6,7].

B-cell lymphoma/leukemia 10 (Bcl10) is an adaptor protein associated with the constitutive activation of canonical NF-κB in mucosa-associated lymphoid tissue (MALT) B cell lymphoma [8]. Bcl10 has been largely studied in antigen receptor-mediated lymphocyte activation, where it was found to interact with CARMA/CARD-containing scaffold proteins and MALT1 paracaspase to form the three-component CBM signalosome that sets in motion a cascade of events that eventually lead to NF-κB induction [9,10]. Depending on the cell-specific chromatin landscape accessible to NF-κB or other transcription factors, the activation of this pathway through CBM complex signaling results in the inducible expression of numerous inflammatory cytokines, chemokines and factors that control cellular functions, including survival, proliferation and differentiation [11]. Thus, CBM signaling is essential for host defense and tissue homeostasis. However, alterations in the signaling components, including Bcl10, have been implicated in diseases such as autoinflammatory diseases, lymphoproliferative disorders and immunodeficiencies, as well as cancers [12].

Bcl10 signaling plays an important role in establishing the inflammatory environment associated with asthma. Bcl10 is involved in FcεRI-mediated NF-κB activation, pro-inflammatory cytokine release and degranulation of mast cells [13]. IgE-mediated allergic inflammatory response was found to be impaired in Bcl10-deficient mast cells [13,14], indicating Bcl10 as a key regulator of immune signaling in mast cells. Furthermore, Bcl10 also complexes with CARMA3/CARD10 and MALT1 to form the CBM-3 signalosome as a result of G-protein-coupled receptor (GPCR) activation to induce pro-inflammatory gene expression in non-immune cells such as fibroblasts and endothelial cells [15]. 

Interestingly, studies also indicate Bcl10 to be an important mediator of fibrotic remodeling. Angiotensin II (Ang II)-induced fibrotic signaling, such as extracellular matrix (ECM) synthesis and myofibroblast proliferation, was mediated by Bcl10 via the CBM-3 signalosome [16,17]. Bcl10 also regulated lipopolysaccharide (LPS)-induced activation of NF-κB and IL-8 in human intestinal epithelial cells [18], suggesting a role for Bcl10 in epithelial inflammation. Moreover, NF-κB-regulated genes in the airway epithelium contributed to allergen-induced peribronchial fibrosis and mucus production [19].

NF-κB activation is increasingly recognized in the pathogenesis of asthma and Bcl10-mediated NF-κB signaling has widely been studied as an inflammatory pathway in immune cells [20]. However, the role of Bcl10-driven NF-κB activation is yet to be explored in the context of airway remodeling in asthma. Since Bcl10 is an important mediator of fibrotic remodeling in other disease models, we hypothesized that the Bcl10-mediated NF-κB pathway promotes fibrotic signaling in severe asthmatic fibroblasts. 

## 2. Materials and Methods

### 2.1. Cell Culture and Treatment of Primary Human Bronchial Fibroblasts

The primary bronchial fibroblasts were isolated from endobronchial tissue biopsies obtained from patients with severe asthma or healthy volunteers. These fibroblasts were archived at Quebec Respiratory Health Research Network Tissue Bank (McGill University Health Centre (MUHC)/Meakins-Christie Laboratories Tissue Bank, Montreal, QC, Canada), as previously described [21]. The original study was approved by the MUHC Research Ethics Board (REB) with reference number 2003–1879 and the subjects had provided written informed consent in accordance with the Declaration of Helsinki. The fibroblasts were age-matched with a mean age of 43.4 ± 8.3 years for the severe asthmatics and 43.7 ± 12.5 years for the non-asthmatics. Furthermore, cells were chosen from subjects who were non-smokers, as smoking patients with severe asthma demonstrated a distinct gene expression profile compared to non-smoking severe asthmatics and non-smoking individuals without asthma [22]. 

The cells were revived in Dulbecco’s modified Eagle’s medium (DMEM)—high glucose supplemented with 10% fetal bovine serum (FBS), 2 mM L-glutamine, 100 units/mL of penicillin, and 100 ng/mL streptomycin in 75-cm^2^ flasks. The cells were maintained at 37 °C in 5% CO_2_ with medium change performed every 2–3 days. They were harvested when 80–90% confluent using 0.1% trypsin-ethylenediaminetetraacetic acid (EDTA) solution and seeded into multiwell tissue culture plates for experiments. 

The cells were seeded into 6- or 12-well plates at approximate cell density/well of 10 × 10^4^, and 5 × 10^4^, respectively. At ~70% confluency, they were serum-starved in FBS-free DMEM complete medium for a period of 24 h before experiments. For baseline measurements, the cells were cultured in DMEM complete medium thereafter. For LPS stimulation experiments, the cells were stimulated with 10 µg/mL of LPS (Sigma-Aldrich, Burlington, MA, USA, Cat. No. L8643) for the specified amount of time. Selective inhibition of Interleukin 1b Receptor-associated Kinase (IRAK) 1/4 was carried out to block the Bcl10 upstream signaling. Accordingly, the cells were pre-treated with 50 µM of IRAK 1/4 Inhibitor I (R&D Systems, Minneapolis, MN, USA, Cat. No. 5665/50) for 1 h prior to LPS stimulation for 2 h. 

### 2.2. Quantitative Real-Time Polymerase Chain Reaction (qRT-PCR)

In order to investigate the mRNA expression of the different components of the NF-κB activation pathway and fibrotic markers in bronchial fibroblasts, qRT-PCR was performed. Total RNA was extracted from cell pellets using Trizol Reagent (Invitrogen, Waltham, MA, USA, Cat. No. 15596018), according to manufacturer instructions. RNA quality and yield were determined by Nanodrop (Thermo Scientific, Waltham, MA, USA) spectrophotometric measurements. cDNA synthesis was performed from 200 ng of total RNA using the High-Capacity cDNA Reverse Transcription Kit (Applied Biosystems, Waltham, MA, USA, Cat. No. 4368814) in the Veriti Thermal Cycler (Applied Biosystems, Waltham, MA, USA). qRT-PCR reactions were set up using the 5× Hot FirePol EvaGreen qRT-PCR SuperMix (Solis Biodyne, Tartu, Estonia, Cat. No. 08-36-00001) in QuantStudio 3 Real-Time PCR System (Applied Biosystems, Waltham, MA, USA). The primers used are listed in Table 1. Gene expression was analyzed using the Comparative CT (ΔΔCT) method after normalization to the housekeeping gene 18s RNA. All results are presented as fold expression change compared to non-asthmatic healthy controls in baseline experiments or untreated controls in LPS stimulation experiments.

### 2.3. Western Blotting

In order to investigate the protein expression of Bcl10, western blotting was performed. The cell pellets were lysed in 10X RIPA buffer (abcam, Cambridge, UK, Cat. No. ab156034) diluted to 1X and supplemented with 1× Protease Inhibitor Cocktail (Sigma-Aldrich, Burlington, MA, USA, Cat. No. P2714) and 1 mM PMSF (Sigma-Aldrich, Burlington, MA, USA, Cat. No. P7626). The protein lysates were quantified using the Protein Assay Kit II (Bio-Rad, Hercules, CA, USA, Cat. No. 5000002) with bovine serum albumin (BSA) as standard. The lysates were boiled in 10× Laemmli Sample Buffer diluted to 1×, and total protein was separated using 4–20% Mini-PROTEAN TGX Precast Protein Gels (Biorad, Hercules, CA, USA, Cat. No. 4561095-6). Post electrophoresis, the proteins were transferred onto a 0.2 μm nitrocellulose membrane (Bio-Rad, Hercules, CA, USA, Cat. No. 1620112) and blocked in 5% non-fat dry milk for at least an hour at room temperature (RT) before incubating the membrane overnight at 4 °C with anti-Bcl10 antibody (331.3) (Santa Cruz, TX, USA, Cat. No. sc-5273). The membrane was subsequently incubated for 1 h at RT with the respective horseradish peroxidase (HRP)-linked secondary antibody. The blots were developed using the Clarity™ Western ECL Substrate (Bio-Rad, Hercules, CA, USA, Cat. No. 170-5060) in the ChemiDoc™ Touch Gel and Western Blot Imaging System (Bio-Rad, Hercules, CA, USA). Anti-β-actin (Sigma-Aldrich, Burlington, MA, USA, Cat. No. Cat# A5441) was used as the loading control. Image Lab software (Bio-Rad, Hercules, CA, USA) was used to detect and quantify the protein bands. Protein levels were normalized to β-actin and thereafter to healthy controls.

### 2.4. Immunohistochemistry

Formalin-fixed paraffin-embedded (FFPE) sections of bronchial biopsy tissues on slides were obtained from non-asthmatic control individuals, mild asthma, moderate asthma and severe asthma subjects. These slides were archived at the Biobank of the Quebec Respiratory Health Research Network Canada with MUHC REB number BMB-02-039-t [23]. According to retrieved data, patients had been classified into mild, moderate and severe based on frequency of exacerbations, lung function and medication usage, as previously described [23]. Immunohistochemical staining was performed to determine the expression and distribution of Bcl10 as previously described [24]. Briefly, routine deparaffinization in xylene and rehydration steps in decreasing concentrations of ethanol were performed. Heat-activated antigen retrieval was carried out using Tris EDTA buffer at pH 9.0 for Bcl10, as per manufacturer recommendations. The sections were incubated in hydrogen peroxidase blocking solution for 30 min to block the endogenous peroxidase activity. The slides were then blocked in 1% BSA for 20 min at RT and immunostained using mouse monoclonal anti-Bcl10 (Santa Cruz, TX, USA, Cat. No. sc-5273) antibody overnight at 4 °C. The slides were developed using HRP/DAB (ABC) Detection IHC kit (Abcam, Cambridge, UK, Cat. No. ab64264), according to manufacturer recommendations. The primary antibody was omitted to serve as technical negative control and appropriate positive control tissue was used. Nuclei were counterstained blue with hematoxylin (Thermo Scientific Shandon, Waltham, MA, USA). The stained slides were then examined and analyzed by a histopathologist.

### 2.5. Paraffin Embedding and Immunocytochemistry of Human Bronchial Cells

Human bronchial fibroblasts (HBFs) and human bronchial epithelial cells (HBE) were clotted, processed and immunostained. Briefly, 5 × 10^5^ cells were centrifuged at 1200 rpm for 5 min and placed on ice immediately after centrifugation. 120 μL of plasma was then added dropwise to the cell pellet, followed by gentle vortexing for 10 s. 80 μL of thrombin (Sigma-Aldrich, Burlington, MA, USA, Cat. No. T4393) was then added and mixed so as to clot the cells together. The clotted cells were then transferred to a sheet of Speci-wrap paper, which was folded and secured inside a formalin cassette. The clotted cells were then processed using Excelsior AS Tissue Processor (Thermo Fisher Scientific, Waltham, MA, USA) to generate FFPE blocks of the clotted cells, which were then immunostained as described above.

### 2.6. Statistical Analysis

All data are presented as mean ± standard error of the mean (SEM) of 2–4 independent experiments using GraphPad Prism 6.0 software (GraphPad, San Diego, CA, USA). Data analyses were performed using unpaired two-tailed Student’s *t*-test while comparing NHBF and DHBF, one-way ANOVA followed by Tukey’s multiple comparison tests or two-way ANOVA followed by Sidak’s multiple comparison tests for statistical analysis of the data. A *p* value < 0.05 was considered statistically significant.

## 3. Results

### 3.1. Activation of Bcl10-Mediated NF-κB in Severe Asthmatic Fibroblasts

Persistent activation of the NF-κB signaling pathway characterized the peripheral blood mononuclear cells isolated from patients with severe asthma [3]. We therefore first examined the expression of various components of this pathway in bronchial fibroblasts from severe asthmatics (S-As) and healthy subjects. To compare the basal expression levels, normal human bronchial fibroblasts (NHBF) and asthmatic diseased human bronchial fibroblasts (DHBF) in culture were pelleted and lysed for qRT-PCR analysis. The mRNA expression of TLR4 (*p* = 0.0056), MALT1 (*p* = 0.0006) and CARMA3 (*p* = 0.0067) was significantly upregulated in DHBF relative to NHBF (Figure 1A). IKBα gene expression was significantly downregulated in DHBF (*p* = 0.0116). Furthermore, A20 deubiquitinase, another negative regulator of NF-κB that terminates downstream signaling events [25], was also lowered in DHBF compared to NHBF (Figure 1A). Increased mRNA levels of RELA subunit were also found in DHBF compared to NHBF, albeit without any statistical significance (*p* = 0.09). Since increased expression of the intermediates of the NF-κB pathway was noted, we next measured the expression of NF-κB-target genes, including IL-6 and IL-8. While a significant increase in IL-8 expression was observed in DHBF (*p* = 0.0231), IL-6 transcript levels showed an increased trend in DHBF when compared to NHBF. Bcl10 being a critical mediator of NF-κB signaling, we next investigated the protein expression of Bcl10 in these fibroblasts at basal levels. It was interesting to note a trend of 3-fold increase in the relative protein expression of Bcl10 in DHBF in comparison to NHBF (*p* = 0.0524) (Figure 1B).

The elevated expression of Bcl10 in bronchial fibroblasts from severe asthmatic subjects and the increased signature of key NF-κB genes at baseline suggest activation of the Bcl10-mediated NF-κB pathway in S-As fibroblasts.

### 3.2. Increased Cytoplasmic Bcl10 Expression in Subepithelial Fibroblasts from Severe Asthma Patients

In order to validate the increased protein expression of Bcl10 in severe asthma, we evaluated its immunohistochemical expression and distribution in bronchial biopsies obtained from normal individuals and asthma patients of varying severities. Bcl10 protein was variably expressed in the airways of both normal and asthma patients.

Expression of Bcl10 was strong and more pronounced in the subepithelial fibroblasts in severe asthmatic patients compared to weak to moderate expression noted in mild and moderate asthmatics, respectively (Figure 2C–F). The non-asthmatic healthy individuals showed almost no expression of Bcl10 in subepithelial fibroblasts (Figure 2A,B). On the other hand, the mucosal bronchial epithelial cells showed variable nuclear and cytoplasmic staining for Bcl10 in both healthy and asthmatic patients. 

In the control biopsies, the surface epithelium showed positive Bcl10 expression in both the cytoplasm and the nuclei whereas the subepithelial fibroblasts were mostly negative for Bcl10 (Figure 2A,B). Biopsies from mild and moderate cases of asthma displayed moderate Bcl10 expression in both the epithelium and subepithelium with very few subepithelial fibroblasts staining positively for Bcl10 (Figure 2C,D). Biopsies from severe asthmatic subjects showed an increased number of fibroblasts in the subepithelium with intense Bcl10 staining (Figure 2E,F). These findings suggest that the intensity of positive Bcl10 staining as well as the distribution of Bcl10-positive cells increased with increasing severity of asthma.

### 3.3. Subcellular Localization of Bcl10 in Bronchial Fibroblasts and Bronchial Epithelial Cells

Subcellular localization of Bcl10 correlated with the development of MALT lymphoma, with strong nuclear Bcl10 expression seen in MALT lymphomas with t(1;14)(p22;q32) translocation [26]. To understand the potential role of Bcl10 in fibroblast function, we next assessed their subcellular localization in human bronchial fibroblasts (HBF) and bronchial epithelial cells (HBE) from severe asthmatic and healthy subjects. We clotted the HBFs and HBEs, and stained them for Bcl10 using immunocytochemistry. In bronchial fibroblasts, Bcl10 expression was highly intense in DHBF when compared to NHBF (Figure 3A). Strong and more pronounced cytoplasmic Bcl10 expression was noted in DHBF compared to NHBF. Similarly, in bronchial epithelial cells, a greater number of Bcl10-positive cells were detected in DHBE relative to NHBE (Figure 3B). Taken together, these findings confirm constitutively higher Bcl10 expression in asthma, particularly in severe asthma, and its localization predominantly in the cytoplasmic compartment in bronchial fibroblasts.

### 3.4. Bcl10 Mediates LPS-Induced Pro-Fibrotic Cytokine Signaling in Bronchial Fibroblasts

In order to assess the role of Bcl10 in pro-fibrotic signaling, the bronchial fibroblasts were stimulated with LPS and the level of pro-fibrotic cytokines, IL-6, IL-8 and TGFβ1, was analyzed. The fibroblasts were serum-starved for 24 h and then exposed to 10 µg/mL of LPS for different time-points. We then studied the effect of LPS stimulation on the gene expression of BCL10 using qRT-PCR. LPS stimulation was observed to significantly upregulate the expression of BCL10 in both NHBF and DHBF in comparison to their respective unexposed controls and this increase was found to be time-dependent (Figure 4A). 

Since cross-linking of Toll-like receptor 4 (TLR4) with LPS induces the synthesis and secretion of pro-inflammatory and pro-fibrotic cytokines [27], we next examined the expression of IL-6, IL-8 and TGFβ1 in NHBF and DHBF upon LPS stimulation for 2 h. LPS strongly induced IL-6 gene expression in both NHBF and DHBF, whereas IL-8 and TGFβ1 was induced to a lower extent when compared to their unexposed control (Figure 4C). The induction of collagen expression by LPS in primary cultured mouse lung fibroblast [28] led us to next assess the expression of collagen markers, COL1A1 and COL5A1, and fibronectin in NHBF and DHBF. No significant alteration in the expression of these markers was observed in NHBF and DHBF upon LPS stimulation for 2 h (Figure 4D). 

To further confirm the role of Bcl10 in LPS-induced pro-fibrotic cytokine signaling, we blocked the activity of IRAK1 and IRAK4, upstream adaptors that are responsible for recruiting Bcl10 to the TLR4 signaling complex and further signaling to NF-κB [29] using an IRAK1/4 inhibitor. In addition to restoring Bcl10 levels, the LPS-induced upregulation of IL-6, IL-8 and TGFβ1 was reversed upon exposure to the IRAK inhibitor (Figure 4B,C). The IRAK inhibitor suppressed the elevated gene expression of Bcl10 and IL-6 in both NHBF and DHBF. Interestingly, a reduction in COL5A1 expression and a reducing trend in COL1A1 and FN1 expression was noted in LPS-stimulated NHBF in the presence of IRAK1/4 inhibition (Figure 4D). 

TLR4 can signal via the MyD88-dependent as well as the TRIF-dependent pathways, both of which, in turn, activate the canonical NF-κB pathway [30]. Bcl10-mediated signaling is a part of the MyD88-dependent pathway, which represents the early phase of activation of NF-κB. The TRIF-dependent pathway involves signaling through TBK1 and IRF complexes culminating in IFN-β expression and represents the late phase activation of NF-κB. Since TBK1 was reported to stimulate NF-κB and mediate the induction of inflammatory cytokine genes [31,32], we examined the expression of intermediates of the TRIF-dependent pathway upon stimulation with LPS and in the presence of IRAK1/4 inhibition. IRAK1/4 inhibition disrupts Bcl10 signaling, and therefore, Bcl10-mediated NF-κB signaling, but not TBK1-mediated NF-κB signaling. Therefore, as expected, LPS induced the expression of TBK1, IRF3, IRF7 and IFN-β in both NHBF and DHBF, with a statistically significant increase noted in IRF3 (Figure 5). IRAK1/4 inhibition led to a further increase in IRF3 expression but the expression of TBK1, IRF7 and IFNβ was unaffected. The inhibition of LPS-induced expression of IL-6, IL-8 and TGFβ1 upon inhibiting Bcl10 signaling and not TRIF or TBK1 thus confirms that Bcl10-mediated signaling regulates pro-fibrotic cytokine expression in bronchial fibroblasts.

## 4. Discussion

Human studies as well as in vivo animal models have reported increased activation of the classical and alternative NF-κB pathways in asthmatic airway tissues and in inflammatory cells [3,33]. As such, NF-κB signaling intermediates are attractive therapeutic targets for airway diseases such as asthma, as the underlying inflammation is independent of stimuli [34] and is mediated at least in part by NF-κB mediated signaling in bronchial fibroblasts (Figure 1). Bcl10 being a critical mediator of NF-κB signaling prompted us to explore its role in fibrotic remodeling in bronchial fibroblasts from severe asthma. To the best of our knowledge, this is the first report providing evidence of elevated protein expression of Bcl10 in the pathogenesis of severe asthma as well as the role of Bcl10-mediated signaling in the LPS-induced pro-fibrotic cytokine expression in bronchial fibroblasts. 

NF-κB is a key component of the inflammatory network that controls cytokine production in airway pathology [20]. Overexpression of Bcl10 is an indicator of constitutive NF-κB activation in tumors including MALT lymphoma [8,35], and persistent NF-κB activation is known to characterize severe uncontrolled asthma [3]. At baseline, the S-As fibroblasts demonstrated differential gene expression of various intermediates of the NF-κB pathway when compared to their healthy counterparts, supporting the notion of activation of NF-κB in severe asthma (Figure 1A). This was further confirmed by the increased protein expression of Bcl10 at basal levels in S-As fibroblasts (Figure 1B). We have also previously reported the increased expression of pro-fibrotic and pro-inflammatory mediators associated with the NF-κB activation, such as IL-6, IL-8, IL-11 and GROα (CXCL1) in these S-As fibroblasts [36]. Thus, the increased Bcl10 expression in S-As fibroblasts appears to signify the activation of the Bcl10-mediated NF-κB pathway in severe asthma. The observation that complete Bcl10 deficiency severely impaired fibroblast function in an immunodeficient individual [37] is testament to its importance in fibroblast response. Numerous lymphoid malignancies are characterized by the constitutive aberrant activation of the NF-κB pro-inflammatory pathway. Here, we demonstrated a similar pattern of Bcl10-mediated NF-κB activation in airway structural fibroblasts. 

Furthermore, Bcl10 was differentially expressed in the sub-epithelial fibroblasts among the varying severities of asthma, ranging from weak expression in control biopsies to moderate expression in mild-to-moderate asthma and strong expression in severe asthma (Figure 2A–F). Just as in the case of MALT lymphoma, high Bcl10 expression in fibrotic airway tissues is paradoxical, considering that Bcl10 is a pro-apoptotic CARD-containing adaptor molecule [38]. However, certain cellular contexts in vivo may influence Bcl10 to behave as an anti-apoptotic molecule. For instance, overexpression of Bcl10 conferred a survival advantage to activated primary B cells even after withdrawal of the activating stimuli [39]. Alternately, the subcellular localization of Bcl10 may be a pre-determining factor in explaining this paradox. Bcl10 was predominantly expressed in the cytoplasm of subepithelial fibroblasts irrespective of the disease severity (Figure 2). However, Bcl10 nuclear expression was detected in the bronchial epithelium of non-asthmatic and asthmatic individuals (Figure 2). The different subcellular localization pattern of Bcl10 between the bronchial fibroblasts and epithelial cells (Figure 3A,B), indicates cell-dependent functional role of Bcl10. Here, it is interesting to note that the NF-κB-independent functions of Bcl10 include actin remodeling. For instance, Bcl10 regulates TCR-induced actin polymerization and cell spreading in T cells, and FcγR-induced actin polymerization and phagocytosis in monocytes/macrophages [40]. Subsequently, the role of Bcl10 in actin dynamics, cytoskeletal and membrane remodeling in macrophages was found to entail phagosome formation [41]. This may perhaps explain the cytoplasmic expression of Bcl10 in S-As fibroblasts indicating Bcl10-dependent actin polymerization in addition to its role in NF-κB activation. Actin dynamics and polymerization are key to the contractile property of fibroblasts [42]. Thus, the presence of Bcl10 in the cytoplasmic compartment of S-As fibroblasts may signify its role in regulating actin dynamics and contraction of bronchial fibroblasts, and by this means, contributing to airway hyperresponsiveness in severe asthma. However, further studies are required to completely understand the physiological relevance of this subcellular localization.

Toll-like receptors (TLRs) such as TLR4 and TLR2 are important for the adaptive Th2-cytokine-driven inflammatory response in asthma [43,44]. Engagement of the TLR initiates the recruitment and activation of several adaptor molecules resulting in the activation of multiple signaling cascades, including NF-κB. TLR activation of the NF-κB pathway regulates the expression of immunomodulatory and inflammatory mediators. For instance, LPS binding to the TLR4 initiates an inflammatory cascade that climaxes in the release of cytokines such as IL-8 and IL-6, and the recruitment of an inflammatory infiltrate of lymphocytes, macrophages and polymorphonuclear leukocytes [45,46]. However, the signal transduction of the TLR4-Bcl10-NF-κB axis in asthmatic fibroblasts is far from understood, and the role of Bcl10 in TLR4-mediated fibroblast function is largely unknown. 

Since the basal expression of TLR4, CARMA3, BCL10 and MALT1 was elevated in DHBF when compared to NHBF (Figure 1), we speculated that the TLR4-BCL10-NF-κB axis responds to LPS stimulation in bronchial fibroblasts and stays upregulated in S-As fibroblasts contributing to the activation of NF-κB in these cells. Here, we show that Bcl10 is a mediator of LPS-induced increase in pro-fibrotic cytokine expression (Figure 4). While LPS stimulation for 2 h boosted the expression of Bcl10, IL-6, IL-8 and TGF-β1 in bronchial fibroblasts, IRAK1/4 inhibition reversed the LPS-induced increase in all components, indicating increased Bcl10-mediated signaling contributed to pro-fibrotic cytokine expression upon LPS exposure in bronchial fibroblasts. Although LPS stimulation was previously reported to induce collagen expression in lung fibroblasts [28], there was no increase in COL1A1, COL5A1 and FN1 expression in NHBF and DHBF with LPS stimulation (Figure 4D). This could be attributed to the short exposure to LPS. While 2 h of LPS stimulation was sufficient to induce pro-fibrotic cytokine signaling in these fibroblasts, longer exposure may pave way to increased ECM secretion via activation of phosphoinositide3-kinase-Akt (PI3K-Akt) pathway [28] as well as autocrine signaling from IL-6 and TGF-β1 [47,48]. 

Since TLR4 signaling activates the canonical NF-κB pathway via the MyD88-dependent as well as the TRIF-dependent pathways [30], we next aimed to verify which of these pathways contributed to pro-fibrotic signaling. While IRAK1/4 inhibition had no effect on the TRIF-dependent pathway, the LPS-induced increase in Bcl10 was abolished and this was accompanied by a corresponding decrease in IL-6, IL-8 and TGF-β1 expression (Figure 5). This indicates Bcl10 signaling directly to its downstream mediators, including IL-6, IL-8 and TGF-β1 in response to LPS in bronchial fibroblasts. Furthermore, since Bcl10 inhibition was not accompanied by alterations in the TRIF-dependent induction of type I interferons, anti-viral signaling and immune defense against viral infections will be sustained in these cells. Considering the activated status of NF-κB pathway in S-As fibroblasts, an exaggerated response to LPS may cause greater extent of inflammatory damage and remodeling changes in severe asthmatic airways.

Increased levels of IL-6 were detected in the sputum of asthmatic subjects when compared to healthy controls, which correlated with impaired lung function in allergic asthma [49]. Further, high circulating levels of IL-6 increased the risk of exacerbations by 10% for each 1-pg/μL increase in baseline IL-6 level [50]. IL-6 is a pleiotropic cytokine with both pro-inflammatory and pro-fibrotic functions [51]. Severe asthma is also characterized by elevated IL-8 levels and associated neutrophilia [52,53]. We have previously reported the upregulation of IL-6 and IL-8 expression at basal levels in S-As fibroblasts when compared to non-asthmatic fibroblasts [36]. The ability of these fibroblasts to produce increased levels of ECM proteins as well as pro-inflammatory and pro-fibrotic cytokines have implicated them in remodeling and inflammation, two key processes involved in the pathogenesis of asthma. IL-8 is known to contribute to the pathogenesis of severe asthma by facilitating various features of airway remodeling, including neutrophil recruitment, epithelial-to-mesenchymal transition [54], angiogenesis [55] and proliferation and migration of ASM cells [56]. TGF-β is the central mediator of fibrotic tissue remodeling in asthma [57]. TGF-β gene polymorphisms were recently reported as a risk factor for asthma control [58]. The persistently high levels of TGF-β in severe asthma may contribute to increased collagen secretion from severe asthmatic fibroblasts despite treatment with oral corticosteroids [59]. Here, we showed that Bcl10 mediated the LPS-induced expression of IL-6, IL-8 and TGF-β1 in bronchial fibroblasts, highlighting the pathogenic role of Bcl10-mediated signaling in promoting airway remodeling in severe asthma.

Although extensively studied in immune cells, Bcl10-mediated NF-κB activation is emerging as an important pathway in non-immune cells as well. For instance, angiotensin II promotes liver fibrosis by activating the CBM-3-dependent NF-κB pathway in hepatocytes [16]. Lysophosphatidic acid-induced NF-κB activation and IL-6 production in murine embryonic fibroblasts involves signaling through adapter proteins Bcl10 and Malt1 [60]. In one study, Bcl10 was found to be an essential component of TLR4 response in human primary fibroblasts [37] and Bcl10 deficiency was found to abolish TLR4 signaling in response to LPS stimuli and subsequent production of IL-6 and IL-8. Our results are consistent with the observations made in this study. 

We identified the Bcl10-mediated NF-κB pathway as a mechanism contributing to fibrotic remodeling and inflammation in severe asthma (Figure 6). Further studies are, however, essential to delineate the molecular interactions of Bcl10 to develop a more complete understanding to explain the signal transduction in bronchial fibroblasts. Another interesting option is to explore the kinetics of CBM-3 formation in S-As fibroblasts taking into account the activation of the Bcl10-mediated NF-κB pathway in these fibroblasts.

Some of the limitations of our study include the lack of specific Bcl10 inhibition that would demonstrate the direct causal relationship between BCL10 and NF-κB pathway activation and could be addressed by using BCL10 siRNA/CRISPR and NF-κB dual luciferase reporter assay. However, we were able to show that an IRAK1/4 selective inhibitor that inhibits the signaling upstream of Bcl10 abrogated the pro-fibrotic signaling induced by LPS. Since the S-As fibroblasts were derived from patients on medications such as glucocorticosteroids and biologics, the results need to be interpreted with caution, as these medications are known to alter inflammatory cell signaling events in asthmatic airways [61]. Nevertheless, the enhanced signature of key NF-κB genes at baseline in the S-As fibroblasts suggest refractoriness to steroids in these patients. Another shortcoming of our study is the lack of in vivo investigation that we aim to explore in future studies.

## 5. Conclusions

Although NF-κB signaling is a well-characterized pathway in airway inflammation, we, for the first time, show activation of Bcl10-mediated NF-κB pathway in severe asthmatic fibroblasts. Furthermore, Bcl10 immunoreactivity in the subepithelium of airway tissues varied in intensity and distribution with increasing severity of asthma, relaying its importance in asthma severity. We further identified Bcl10 as a mediator of signal transduction from TLR4 to the activation of NF-κB inducible genes, including of IL-6, IL-8 and TGF-β1, in bronchial fibroblasts. This allows us to selectively target Bcl10 to impede its pathological signaling in severe asthma. Current asthma medications, including corticosteroid therapy, are not completely effective against airway remodeling, necessitating the identification of new therapeutic targets. Targeted therapy aimed at targeting key molecular signaling pathways is emerging as a novel strategy to treat asthma [62]. Bcl10 may serve as a potential biomarker for testing the activation of NF-kB pathway in asthmatic patients and for asthma targeted therapy, taking into account its role in both airway inflammation and remodeling. 

## Figures and Tables

**Figure 1 biomedicines-10-01716-f001:**
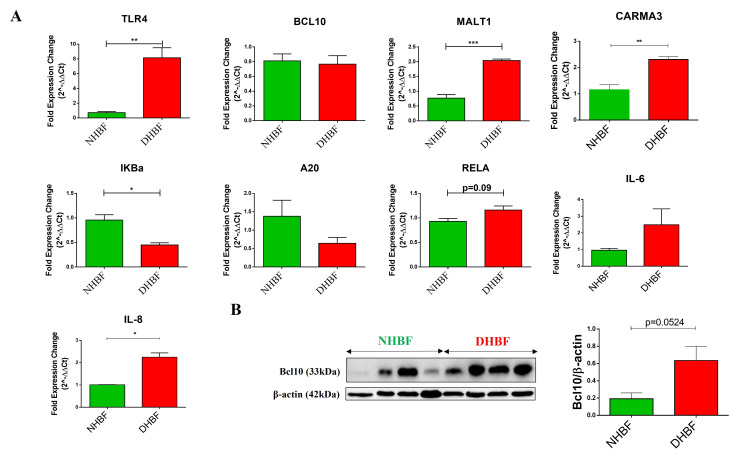
Activation of NF-κB in severe asthmatic fibroblasts. NHBF and DHBF were cultured in DMEM complete medium post serum-starvation. (**A**) Under basal conditions, mRNA expression of NF-κB pathway members, TLR4, BCL10, MALT1, CARMA3, IκBα, A20, RELA, IL-6 and IL-8, in NHBF and DHBF was analyzed by qRT-PCR and expressed as fold expression change relative to NHBF post normalization to housekeeping gene 18s rRNA. (**B**) Whole cell lysates were subjected to immunoblot analysis of Bcl10 protein levels. β-actin was used as loading control. Data are represented as mean ± SEM from at least 3 unique donors in each group. * *p* < 0.05, ** *p* < 0.01, *** *p* < 0.001 determined by unpaired two-tailed Student *t*-test.

**Figure 2 biomedicines-10-01716-f002:**
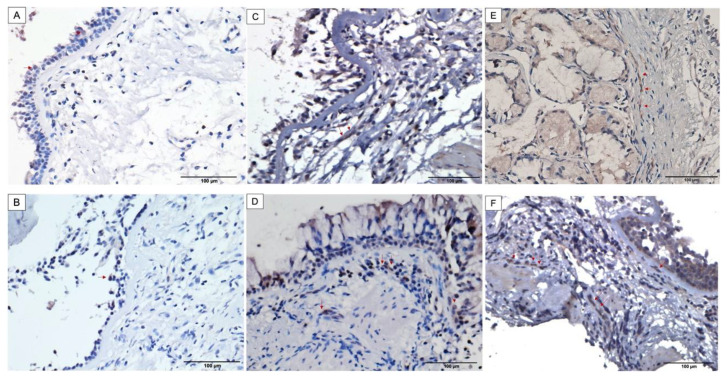
Increased cytoplasmic Bcl10 expression in subepithelial fibroblasts of bronchial biopsies from asthma patients with increasing severity. Representative images taken at 400X magnification showing Bcl10 expression. Representative bronchial biopsy sections from (**A**,**B**) healthy control showing positive Bcl10 staining in surface epithelium and no Bcl10 staining in subepithelial fibroblasts, (**C**,**D**) mild and moderate asthma patients showing minimal and weak Bcl10 cytoplasmic expression in subepithelial fibroblasts, (**E**,**F**) severe asthmatic patients showing numerous subepithelial fibroblasts with strong Bcl10 expression. (Red arrows indicate Bcl10-positive cells).

**Figure 3 biomedicines-10-01716-f003:**
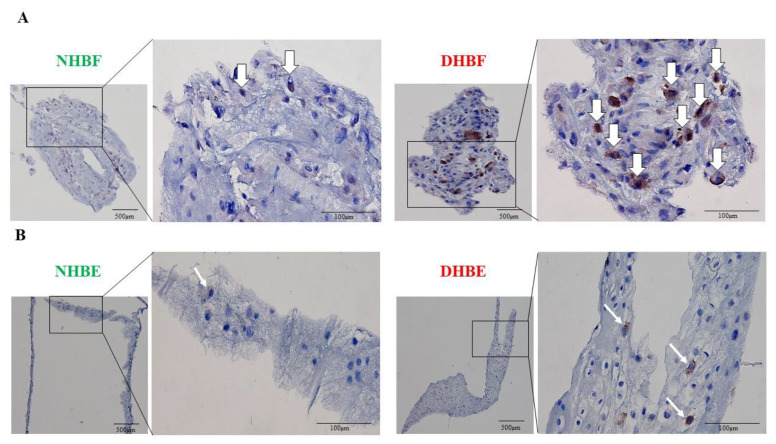
Subcellular localization of Bcl10 in bronchial fibroblasts and bronchial epithelial cells. NHBF and DHBF were cultured in DMEM complete medium, and NHBE and DHBE were cultured in Pneumacult complete medium. (**A**) Cellular clots of NHBF and DHBF, and (**B**) NHBE and DHBE were immunostained for Bcl10 and the white arrows indicate Bcl10-positive cells.

**Figure 4 biomedicines-10-01716-f004:**
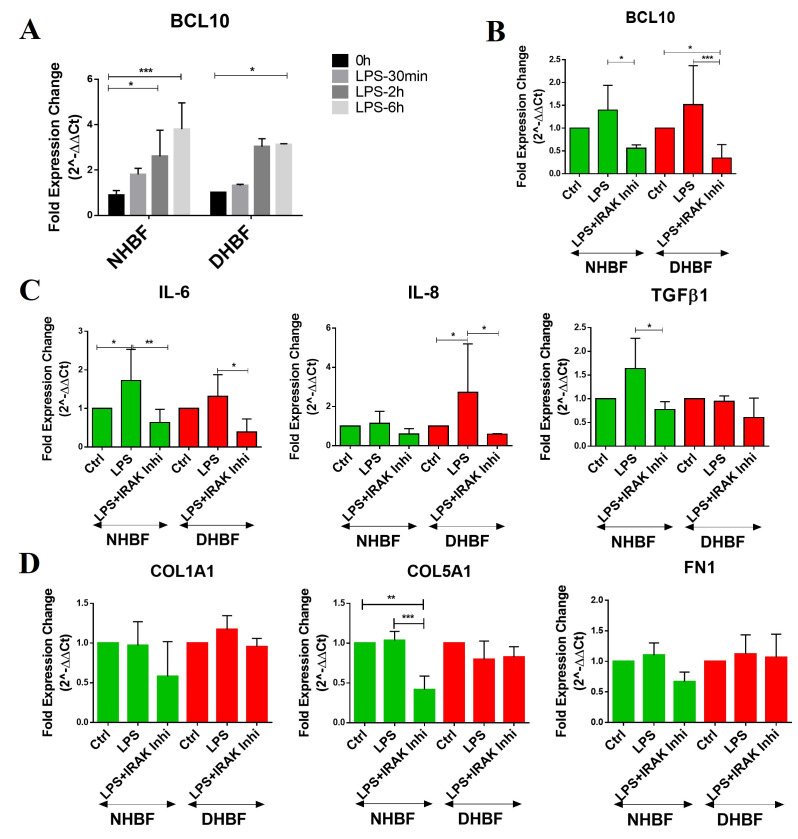
Bcl10 mediates LPS-induced pro-fibrotic cytokine signaling in bronchial fibroblasts. (**A**) NHBF and DHBF were serum-starved for 24 h, and thereafter, exposed to LPS (10 µg/mL) for the indicated time points for mRNA analysis. The effect of LPS treatment on the mRNA expression of BCL10 in NHBF and DHBF was analyzed by qRT-PCR and expressed as fold expression change relative to the respective untreated control at 0 h post normalization to housekeeping gene 18s rRNA. NHBF and DHBF were serum-starved for 24 h, and thereafter, pre-treated with IRAK 1/4 Inhibitor I (50 µM) for 1 h prior to LPS (10 µg/mL) stimulation for 2 h for mRNA analysis. The effect of LPS treatment and IRAK1/4 inhibition on the mRNA expression of (**B**) BCL10, (**C**) pro-fibrotic cytokines, IL-6, IL-8 and TGF-β1, and (**D**) ECM components, COL1A1, COL5A1 and FN1, in NHBF and DHBF was analyzed by qRT-PCR and expressed as fold expression change relative to the respective untreated control post normalization to housekeeping gene 18s rRNA. Data representative of at least 3 independent experiments. Data presented as mean ± SEM after normalization to untreated control. * *p* < 0.05, ** *p* < 0.01, *** *p* < 0.001, statistical significance assessed by one-way ANOVA with Tukey’s multiple comparison tests.

**Figure 5 biomedicines-10-01716-f005:**
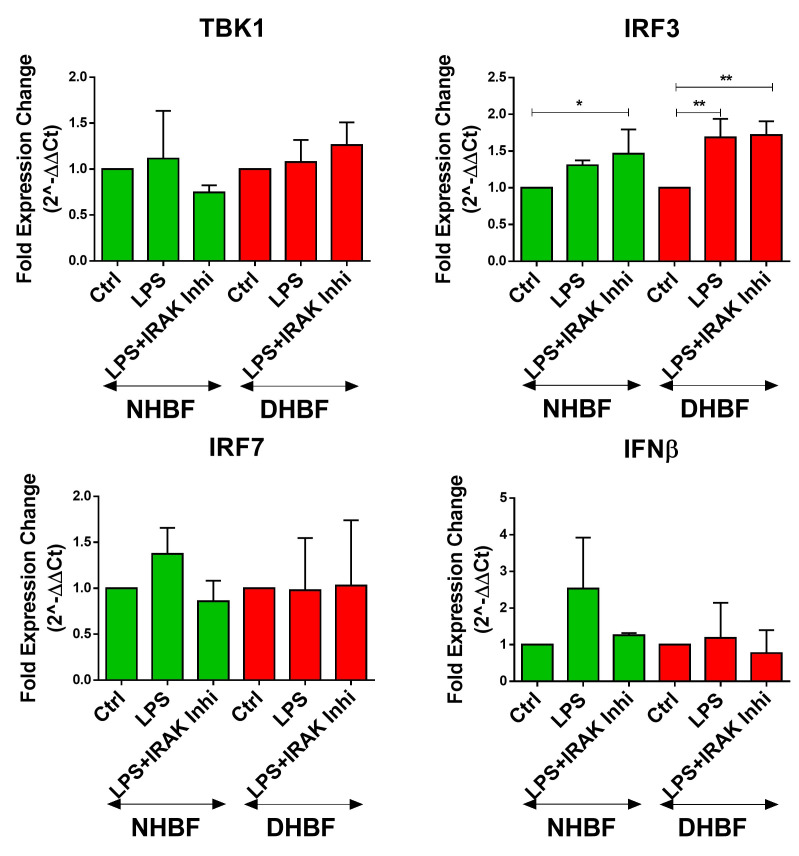
TRIF-dependent pathway not involved in LPS-induced pro-fibrotic cytokine signaling in bronchial fibroblasts. NHBF and DHBF were serum-starved for 24 h, and thereafter, pre-treated with IRAK 1/4 Inhibitor I (50 µM) for 1 h prior to LPS (10 µg/mL) stimulation for 2 h for mRNA analysis. The effect of LPS treatment and IRAK1/4 inhibition on the mRNA expression of TBK1, IRF3, IRF7 and IFN-β in NHBF and DHBF was analyzed by qRT-PCR and expressed as fold expression change relative to the respective untreated control post normalization to housekeeping gene 18s rRNA. Data presented as mean ± SEM after normalization to untreated control. * *p* < 0.05, ** *p* < 0.01, statistical significance assessed by one-way ANOVA with Tukey’s multiple comparison tests.

**Figure 6 biomedicines-10-01716-f006:**
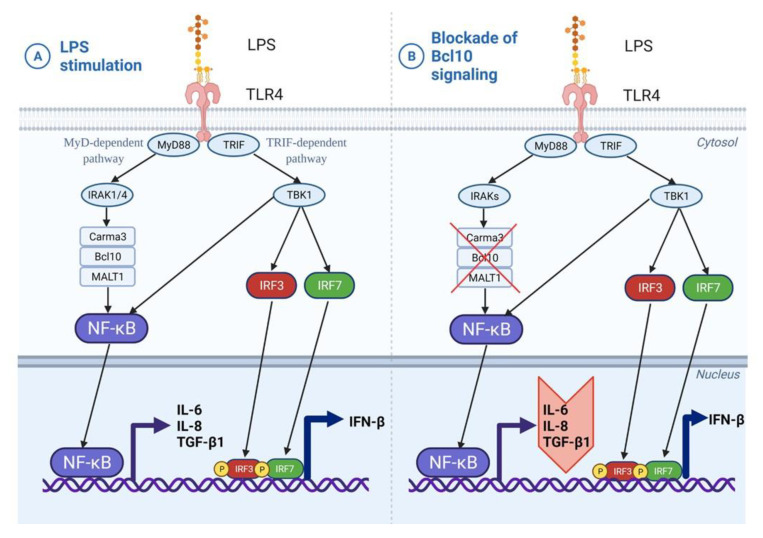
Induction of Bcl10-mediated pro-fibrotic cytokine signaling by LPS in bronchial fibroblasts. (**A**) In bronchial fibroblasts, LPS stimulation of TLR4 receptor activates both the MyD88-dependent and TRIF-dependent pathways. Bcl10-mediated NF-κB activation has downstream effects on airway inflammation and remodeling through the secretion of IL-6, IL-8 and TGF-β1 cytokines. (**B**) Blockade of this Bcl10 signaling cascade may ameliorate both inflammation and fibrosis by impeding the expression of IL-6, IL-8 and TGF-β1 cytokines.

**Table 1 biomedicines-10-01716-t001:** List of primer sequences.

Gene Name	Forward Primer (5′–3′)	Reverse Primer (5′–3′)
TLR4	GCAGTTTCTGAGCAGTCGTGC	CGTCTCCAGAAGATGTGCCGC
BCL10	GAAGTGAAGAAGGACGCCTTAG	AGATGATCAAAATGTCTCTCAGC
MALT1	CTCCGCCTCAGTTGCCTAGA	CAACCTTTTTCACCCATTAACTTCA
CARMA3/CARD10	GGAGCCTCAGACCCTACAGTT	GCAGGTCTAGCAGGTTACGG
IκBα	CTGGGCATCGTGGAGCTTTTGG	TCTGTTGACATCAGCCCCACAC
A20	AATGGCTTCCACAGACACACC	CAAAGGGGCGAAATTGGAACC
RELA	GCCGAGTGAACCGAAACTCTGG	TTGTCGGTGCACATCAGCTTGC
IL-6	GAAAGCAGCAAAGAGGCAC	GCACAGCTCTGGCTTGTTCC
IL-8	CCACACTGCGCCAACACAG	CTTCTCCACAACCCTCTGC
TGF-β1	AAATTGAGGGCTTTCGCCTTA	GAACCCGTTGATGTCCACTTG
COL1A1	GATTGACCCCAACCAAGGCTG	GCCGAACCAGACATGCCTC
COL5A1	GTCGATCCTAACCAAGGATGC	GAACCAGGAGCCCGGGTTTTC
FN1	CTGGGAACACTTACCGAGTGGG	CCACCAGTCTCATGTGGTCTCC
TBK1	AGCGGCAGAGTTAGGTGAAA	CCAGTGATCCACCTGGAGAT
IRF3	TCTGCCCTCAACCGCAAAGAAG	TACTGCCTCCATTGGTGTC
IRF7	GCTGGACGTGACCATCATGTA	GGGCCGTATAGGAACGTGC
IFNβ1	CCTGTGGCAATTGAATGGGAGGC	AGATGGTCAATGCGGCGTCCTC
18S	TGACTCAACACGGGAAACC	TCGCTCCACCAACTAAGAAC

## Data Availability

The data presented in this study are available within the article.

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
