# Peer review of "Bcl10 Regulates Lipopolysaccharide-Induced Pro-Fibrotic Signaling in Bronchial Fibroblasts from Severe Asthma Patients"

_biomedicines, 2022, doi:10.3390/biomedicines10071716_

Round 1

Reviewer 1 Report

The manuscript addresses an important topic of regulation of fibroblasts in severe asthma. The authors placed an innovative idea into focus and conducted an experiment to prove the role of Bcl10 in human primary cell lines obtained from severe asthmatics. Laudably, the authors chose a regulatory pathway with no information in asthma therefore paving a way towards better understanding of subepithelial fibrosis seen in the disease.

It is a carefully written manuscript with clear aims, well built research methodology and logical presentation of the results.

Comments:

For proper understanding and interpretation of data it is important to describe the different groups of samples (asthma severity, treatment in each group, any comorbidity, length of the disease). This is especially important because steroids used in almost all severe asthmatics may have pronounced influence on expression of different genes. Biologics may also have unexpected effects of gene expression. It should be included into the method session and also potential influence of treatment should be discussed.

In line 129 it is not clear what the authors mean under “All results are presented as fold expression change compared to non-asthmatic healthy controls or untreated controls”. Were severe asthmatics untreated or treated severe asthmatics were compared to untreated mild or moderate asthmatics? Did the authors compared the group of samples from severe asthmatic to both groups (health and untreated)? A clear design should be added and a biostatistician should be consulted for the application of appropriate statistical methods.

Please provide information about smokers, ex-smokers, non-smokers. It is published by the collaborators of a major severe asthma study, the U-Biopred that smoking has profound influence on some gene expression. Knowledge gained from these publications should be incorporated into the manuscript and the relevant papers cited.

In line 209 and 209 it is stated that „interesting to note a 3-fold increase in the relative protein expression of 205 Bcl10 in DHBF in comparison to NHBF (p=0.0524)”. As p value shows no significant difference, either more measurements or at least a power calculation should be added and the sentence should be formulated accordingly because now it falsely suggests an „increase” while p demonstrates only a statistical tendency.

Author Response

Reviewer 1

The manuscript addresses an important topic of regulation of fibroblasts in severe asthma. The authors placed an innovative idea into focus and conducted an experiment to prove the role of Bcl10 in human primary cell lines obtained from severe asthmatics. Laudably, the authors chose a regulatory pathway with no information in asthma therefore paving a way towards better understanding of subepithelial fibrosis seen in the disease.

It is a carefully written manuscript with clear aims, well built research methodology and logical presentation of the results.

We would like to thank the reviewer for their comments and time in critically reviewing our manuscript. We hope we have addressed your concerns satisfactorily. Please find our response to each point raised below:

Comments:

For proper understanding and interpretation of data it is important to describe the different groups of samples (asthma severity, treatment in each group, any comorbidity, length of the disease). This is especially important because steroids used in almost all severe asthmatics may have pronounced influence on expression of different genes. Biologics may also have unexpected effects of gene expression. It should be included into the method session and also potential influence of treatment should be discussed.

Thank you for bringing this important point to our attention. Since the primary human fibroblasts use for in vitro work and tissue samples used for IHC were archived at the Quebec Respiratory Health Research Network Tissue Bank, we were able to retrieve the available clinical information. For the in vitro study, fibroblasts from patients with severe asthma and healthy controls were only used. Unfortunately, some of the requested information such as co-morbidity and length of the disease were not recorded for these patients.

We have added the available clinical information to our Methods section (line 98 and line 161). Since patients with severe asthma are usually on corticosteroids for adequate disease management, as was the cohort included in our study, we have discussed the medication use in these patients as a potential limitation of this study (line 521).

In line 129 it is not clear what the authors mean under “All results are presented as fold expression change compared to non-asthmatic healthy controls or untreated controls”. Were severe asthmatics untreated or treated severe asthmatics were compared to untreated mild or moderate asthmatics? Did the authors compared the group of samples from severe asthmatic to both groups (health and untreated)? A clear design should be added and a biostatistician should be consulted for the application of appropriate statistical methods.

We would like to apologize for this confusion. In this study, we have used only fibroblasts from severe asthmatics and healthy controls. At baseline, the severe asthmatic gene expression was compared to non-asthmatic healthy controls (Fig 1). However, in the LPS stimulation experiments, the severe asthmatic and healthy fibroblasts were compared to their respective untreated/unstimulated controls (Fig. 4-5). We have also further clarified this in the figure legends.

Please provide information about smokers, ex-smokers, non-smokers. It is published by the collaborators of a major severe asthma study, the U-Biopred that smoking has profound influence on some gene expression. Knowledge gained from these publications should be incorporated into the manuscript and the relevant papers cited.

Thank you for bringing forward this point as it was an important concern for us. Since LPS inhalation was reported to promote increased alveolar-capillary membrane permeability, exaggerated inflammation, increased epithelial injury and endothelial dysfunction in cigarette smokers when compared to non-smokers (PMID: 26839359), we chose fibroblasts from non-smoking subjects alone for this study to avoid smoking as a confounding factor. We have further clarified this point in the Methods section (line 102).

In line 209, it is stated that “interesting to note a 3-fold increase in the relative protein expression of Bcl10 in DHBF in comparison to NHBF (p=0.0524)”. As p value shows no significant difference, either more measurements or at least a power calculation should be added and the sentence should be formulated accordingly because now it falsely suggests an “increase” while p demonstrates only a statistical tendency.

We have modified this statement to be more factually right (line 214).

Reviewer 2 Report

Authors provided an important information about the asthma pathogenesis and possible lung fibrosis. The manuscript needs some corrections. Plese, provide clear ethic statement as separate section. Lines 105-106,  use an upper index for the degree. All microphotos need mkm scales. 

Concerning the results of immunohistochemistry (Figure 2), more displaying microphotos  are needed. Possibly, you might use higher magnification. 

To our mind, authors need to show the collagen synthesis in fibroblasts (western-blotting).

Please, provide limitation subsection in the discussion.

Author Response

Reviewer 2

Authors provided an important information about the asthma pathogenesis and possible lung fibrosis. The manuscript needs some corrections.

We would like to thank the reviewer for their comments and time in critically reviewing our manuscript. We hope we have addressed your concerns satisfactorily. Please find our response to each point raised below:

Please, provide clear ethic statement as separate section.

Thank you for bringing this to our attention. We have included the ethics statement in the Methodology (line 99 and 161) as well as a separate section (line 560). The samples used in our study were archived at the Biobank of the Quebec Respiratory Health Research Network Canada. Therefore, we have included the Institutional Review Board Statement and approval number of the original study that catered to bronchial biopsy collection and fibroblasts isolation.

Lines 105-106, use an upper index for the degree.

We have made the requested changes.

All microphotos need mkm scales.

As requested, the scales have been added to all images (Fig. 2 and 3).

Concerning the results of immunohistochemistry (Figure 2), more displaying microphotos are needed. Possibly, you might use higher magnification.

All histopathology figures have been changed. The scales were added and all the figures included were phototographed at a high power of 400x.

To our mind, authors need to show the collagen synthesis in fibroblasts (western-blotting).

We appreciate this feedback from the reviewer. However, due to the lack of collagen antibodies in our lab and shortage of time to order new ones, we resorted to analyzing the mRNA expression of collagen markers. Although IRAK inhibition demonstrated a trend of reduction in the gene expression of COL1A1 and FN1 in NHBF, it showed significance only in COL5A1 expression and no significant change was observed in DHBF (Fig. 4D). We believe this could be attributed to the reduced exposure time to the treatments, which was sufficient to alter cytokine expression but not collagen expression. We have discussed this further in the Discussion (line 452).

Please, provide limitation subsection in the discussion.

Thank you for this suggestion. We have added a section on limitations in the Discussion (line 516).

Round 2

Reviewer 2 Report

Authors provided appropriate corrections.